# Chronic Exposure to Nitric Oxide Induces P53 Mutations and Malignant-like Features in Human Breast Epithelial Cells

**DOI:** 10.3390/biom13020311

**Published:** 2023-02-07

**Authors:** Robert Y. S. Cheng, Sandra Burkett, Stefan Ambs, Terry Moody, David A. Wink, Lisa A. Ridnour

**Affiliations:** 1Cancer Innovation Laboratory, Center for Cancer Research, National Cancer Institute, Frederick, MD 21702, USA; 2Molecular Cytogenetics Section, Mouse Cancer Genetics Program, Center for Cancer Research, National Cancer Institute, Frederick, MD 21702, USA; 3Laboratory of Human Carcinogenesis, Center for Cancer Research, National Cancer Institute, Bethesda, MD 20892, USA; 4Center for Cancer Training Office of Training and Education, National Cancer Institute, Bethesda, MD 20892, USA

**Keywords:** nitric oxide, TP53, mutation, breast cancer

## Abstract

The small endogenous signaling molecule nitric oxide (NO) has been linked with chronic inflammation and cancer. The effects of NO are both concentration and temporally dependent; under some conditions, NO protects against damage caused by reactive oxygen species and activates P53 signaling. During chronic inflammation, NO causes DNA damage and inhibits repair proteins. To extend our understanding of the roles of NO during carcinogenesis, we investigated the possible effects of chronic NO exposure on MCF10A breast epithelial cells, as defined by changes in cellular morphology, chromosome/genomic stability, RNA, and protein expression, and altered cell phenotypes. Human MCF10A cells were maintained in varying doses of the NO donor DETANO for three weeks. Distinct patterns of genomic modifications in TP53 and KRAS target genes were detected in NO-treated cells when compared to background mutations. In addition, quantitative real-time PCR demonstrated an increase in the expression of cancer stem cell (CSC) marker CD44 after prolonged exposure to 300 μM DETANO. While similar changes in cell morphology were found in cells exposed to 300–500 μM DETANO, cells cultured in 100 μM DETANO exhibited enhanced motility. In addition, 100 μM NO-treated cells proliferated in serum-free media and selected clonal populations and pooled cells formed colonies in soft agar that were clustered and disorganized. These findings show that chronic exposure to NO generates altered breast epithelial cell phenotypes with malignant characteristics.

## 1. Introduction

The link between chronic inflammation and cancer is multifaceted and well documented [1]. In some conditions, nitric oxide (NO) has been shown to protect against cellular damage by inhibiting the cytotoxicity of reactive oxygen species [2] and eliciting an adaptive activation of P53 via ATM- and ATR-mediated P-Ser-15 and cell cycle checkpoints, as suggested by the presence of P21WAF1 in ulcerative colitis tissue [3]. However, during chronic inflammatory conditions, the sustained production of reactive oxygen and nitrogen species (ROS/RNS) promote damage ranging from DNA mutations to posttranslational modifications of repair proteins [1]. Indeed, NOS2-derived NO has been linked with chronic inflammation as well as increases in both P53 mutation and cancer risk in patients with inflammatory diseases including Crohn’s disease and inflammatory bowel disease [4,5]. In contrast, NOS2 knockout mice demonstrated reduced long-term inflammatory damage to the colon when compared to wild-type mice in a trinitrobenzene-induced chronic colitis model [6]. These observations demonstrate a role of NO in genomic alterations and carcinogenesis.

Elevated tumor NOS2 expression predicts poor disease-specific survival in estrogen receptor negative (ER-) and other aggressive cancers [7,8,9,10,11]. In addition, exogenous NO promotes pro-tumor signaling, increased stem cell biomarker expression, and cell migration and invasion [7,12,13,14]. Also, tumor NOS2 expression correlated positively with tumor vascularization (CD31) and P53 mutations in aggressive ER- breast tumors but not the less aggressive ER+ tumors [7]. Under normal circumstances, many stressors activate wild-type P53, which coordinates the transcription of target genes to promote tumor suppression. In contrast, mutations in the DNA-binding domain of the P53 protein can be pro-oncogenic and have been identified in almost 50% of all cancers [15,16,17]. This observation shows that regulation of transcriptional target genes may play a crucial role in the carcinogenic action of mutant P53 [16]. In addition, mutant p53 oncogenic functions include pro-survival and proliferative signaling, as well as tissue remodeling in relation to migration, invasion, angiogenesis, and stem cell proliferation [16]. Given that wild-type P53 negatively regulates NOS2 [18], and the oncogenic actions of mutant P53 are comparable to mechanisms of NOS2 identified during breast cancer disease progression [7,8,9,12,19,20], this study investigates potential NO-induced P53 mutations and subsequent transcriptional and phenotypic changes associated with chronic exposure of breast epithelial cells to the exogenous NO donor DETANO.

## 2. Materials and Methods

### 2.1. Cell Culture

MCF10A cells (#CRL-10317,) were purchased from ATCC (Manassas, VA, USA) and cultured in MEBM mammary epithelial cell growth medium (#CC-3151) supplemented with bovine pituitary extract, human epidermal growth factor, hydrocortisone, insulin, and gentamicin SingleQuots (#CC-4136) (Lonza, Walkersville, MD, USA) and maintained at 37 °C in an atmosphere of 5% CO_2_ in room air. MCF10A cells were plated at a density of 400,000 cells per 60 mm cell culture dish and grown overnight. The cells were then maintained for three weeks in complete media containing the NO donor DETANO at concentrations of 0, 100, 300, and 500 μM. The media was changed each Monday and Friday and fresh DETANO was added. During this treatment period, the cells were passaged as necessary to avoid confluence. At the end of the three-week treatment, DETANO was removed, and the cells were examined for phenotypic changes.

### 2.2. Cell Growth in Serum-Free Media Supplemented with Selenium, Insulin, and Transferrin (SIT)

The capacity of control and DETANO-cultured cells to grow in serum-free conditions was evaluated. Control and DETANO-treated MCF10A cells were seeded at densities of 100,000 cells per 60 mm tissue culture dish and maintained for three weeks in serum-free RPMI medium supplemented with 5 ng/mL selenium, 5 g/mL insulin, and 5 g/mL transferrin (SIT) [21]. MCF10A cells cultured in 100 μM DETANO produced colonies in the SIT media. Single colonies were ring-isolated, and the remaining cells were pooled for further characterization.

### 2.3. Colony Formation in Soft Agar

Colony formation in soft agar was assessed. Six-well cell culture dishes were prepared with a bottom layer consisting of 0.5% agarose in SIT medium with 5% FBS added. The top layer included 50,000 cells suspended in 3 mL of 0.3% agarose in SIT medium. After two weeks, 1 mL of 0.1% p-iodonitrotetrazolium violet was applied to triplicate wells and incubated at 37 °C for 16 h. The plates were examined for colony formation, and the number of colonies larger than 50 µm in diameter were counted using an Omnicon image analysis system.

### 2.4. Scratch Test

MCF10A cells were seeded into a 60 mm dish and the cells were grown to 100% confluence. A straight line was scratched across the confluent monolayer using a 1000 μL pipette tip. The cells were gently rinsed with PBS to remove detached cells and complete media was added back to each dish. The scratch line was imaged for gap closure at time 0, 24, 48, and 96 h on a AMEX1000 inverted microscope (ThermoFisher, Waltham, MA, USA).

### 2.5. Karyotype Analysis

The metaphases were arrested by incubation with Colcemid (#15210-040, KaryoMax Colcemid Solution (Invitrogen, Carlsbad, CA, USA) (10 ug/mL) 2 h prior to harvest. Cells were collected and treated with hypotonic solution (#685804, KCL 0.075M, Macron Chemical, Radnor, PA, USA) for 15 min at 37 °C and fixed with 3:1 ratio of methanol: acetic acid. Slides were prepared and aged overnight for use in G-band, SKY analysis. Chromosomes were stained with a trypsin- Giemsa staining technique [22]. Analyses were performed under an Axioplan ImagerZ2 (Zeiss, Dublin, CA, USA) microscope coupled with a CCD camera; images were captured with ASI software, (Applied Spectral Imaging Inc., Vista, CA, USA). The karyotype was determined by comparison to the standard ideogram of banding patterns for human chromosomes by ISCN 2009.

### 2.6. Spectral Karyotyping

Spectral karyotyping was performed according to the manufacturer’s protocol using 24-color human SKY paint probes (#FPRPR0028, Applied Spectral Imaging, Carlsbad, CA, USA). Hybridization was carried out in a humidity chamber at 37 °C for 16 h. The post-hybridization rapid wash procedure was used with 0.4X SSC at 72 °C for 4 min, detection was carried out according to manufacturer’s protocol. Spectral images of the hybridized metaphases were acquired using a SD301 SpectraCube system (Applied Spectral Imaging Inc., Carlsbad, CA, USA) mounted on top of an epi-fluorescence microscope Axioplan ImagerZ2 (Zeiss, Dublin, CA, USA). Images were analyzed using Spectral Imaging HiSKY 7.2 acquisition software (Applied Spectral Imaging Inc., Carlsbad, CA, USA). At least 10 SKY hybridized metaphases were analyzed in this experiment. Metaphase spreads were equilibrated in 2 × SSC (30 mM sodium citrate, 300 mM NaCl, pH 7) for ~5 min. Slides were dehydrated using successive washes of 75, 85, and 100% ethanol for 2 min each and allowed to dry. Samples were placed on ThermoBrite (Leica, Teaneck, NJ, USA) for co-denaturation and hybridization, the program was set to denature at 75 °C for 3 min and hybridize 37 °C for overnight in a humid condition (ThermoBrite, Leica, Teaneck, NJ, USA). Samples were washed successively in 0.4 × SSC/0.3% Tween20 at 73 °C for 2 min and 2 × SSC with 0.1% Tween-20 at room temperature for 2 min. Samples were briefly rinsed with H_2_O. Samples were mounted with Prolong Gold Antifade with DAPI (P36935, Invitrogen, Life Technologies, Carlsbad, CA, USA), #1.5 coverslips.

### 2.7. TOPO TA Cloning

Target amplicons covering the hot spots of the following gene exons, TP53 (exon 5, 6, 7 and 8); KRAS (exon 2 and 3), PIK3CA (exon 9 and 20) were examined. The PCR primer pairs (Appendix A) were designed to cover each exon starting from the intronic regions and were verified by melting curve analysis and agarose gel electrophoresis. All samples were amplified with high fidelity Platinum Taq DNA polymerase (Invitrogen, Carlsbad, CA, USA) under the following conditions: 5 min at 95 °C, followed by 40 cycles of 15 s at 95 °C and then 30 s at 60 °C. The amplicons were purified with QIAquick PCR purification kit (Qiagen, Germantown, MD, USA) prior to cloning into the One Shot Top10 competent cells (Invitrogen, Carlsbad, CA, USA) following the manufacturer’s protocol. Ten positive colonies from each amplicon were picked for Sanger sequencing. To minimize sequencing bias, ten colonies were divided into two groups and sequenced from opposite directions. Sequencing data were aligned using the human reference genome (GRCh38) and compared side-by-side with Clustal Omega [23].

### 2.8. RNA Transcript Expression Level Analyses

Total RNA was extracted with TRIzol reagent (Invitrogen, Carlsbad, CA, USA) following the manufacturer protocol. Five microgram total RNA per sample was used for cDNA synthesis. cDNAs were generated with EcoDry cDNA synthesis Kit (Clontech, San Jose, CA, USA) in a 20 μL reaction volume. Quantitative real time PCR was performed to assess the amplification levels; cancer stem cell (CSC) gene primer pairs (Appendix A) were designed using Primer3 online tool (https://primer3.ut.ee/ (accessed on 10 October 2012)). All primer pairs included at least one end that recognized the exon-exon-junction to avoid genomic DNA amplification. All PCRs adhered to universal PCR conditions as follows: 2 min at 95 °C, 40 cycles of 5 s at 95 °C and 31 s at 60 °C. Primer pairs are optimized and examined to ensure similar amplification efficiencies prior to PCR assays. Relative gene expression levels were calculated using the delta-delta-Ct approach [24].

### 2.9. Protein Expression Analyses

Protein concentrations of cell lysates were determined using the BCA protein assay (Thermo Scientific, Rockford, IL, USA). Heat denatured protein from cell lysates were electrophoresed on 4–20% SDS-polyacrylamide gels then transferred onto iBLOT stack nitrocellulose membranes (Invitrogen Carlsbad, CA, USA). Transferred protein was blocked for 1 h at room temperature in 1% milk in T-TBS buffer and then incubated overnight at 4 °C with primary antibodies followed by HRP-conjugated secondary antibodies. The blots were washed and exposed to ECL substrate. Immunoreactive protein was visualized on an Alpha Innotech FluorChem Imager (Cell Biosciences, Santa Clara, CA, USA).

### 2.10. Animal Study

Animal care was provided at the NCI-Frederick Animal Facility according to procedures outlined in the Guide for Care and Use of Laboratory Animals. The NCI-Frederick facility is accredited by the Association for Accreditation of Laboratory Animal Care International and follows the Public Health Service Policy for the Care and Use of Laboratory Animals. Female athymic nude mice obtained from the Frederick Cancer Research and Development Center Animal Production Area were housed five per cage and given autoclaved food and water ad libitum. Eight-week-old female athymic nude mice were briefly anesthetized with isoflurane and then subcutaneous tumor cell injections in the fourth mammary fat pad were performed with 1 × 10^6^ MCF10A control or selected clonal populations that were ring isolated from MCF10A cells cultured in 100 µM DETANO. In addition, MCF10A clonal populations were mixed with MDA-MB-468 cells (MB-468) (2:1) to explore the potential of MCF10A clonal populations to increase MDA-MB-468 growth. Mice were monitored Monday, Wednesday, Friday for tumor growth, which was measured in two directions by caliper. Tumor volume was determined by the following equation: [(short diameter)^2^ × long diameter]/2. Mice were euthanized 30 days after cell injections.

### 2.11. Statistical Analysis

Charts and gene figures were generated in GraphPad Prism 8 (San Diego, CA, USA) and Microsoft PowerPoint (Office 365, Redmond, WA, USA) respectively. Results are presented as mean +/− SEM. Student t-test was used to determine statistical significance as defined by *p* < 0.05.

## 3. Results

### 3.1. Phenotypic Assays

Chronic NO exposure was accomplished by maintaining MCF10A cells in complete media supplemented with increasing concentrations of the exogenous NO donor DETANO (T_1/2_ at 37 °C ~ 20 h) for three weeks as described in the Materials and Methods section. At the end of the three-week period, the DETANO was removed, and the cells were examined for altered physical and genetic characteristics suggestive of tumorigenesis. The most obvious change pertained to altered cellular morphology; while control MCF10A cells showed characteristic cuboidal morphology, cells previously maintained in 100–500 μM DETANO acquired elongated features (Figure 1). While cells treated with 100 μM DETANO were mainly cuboidal with some elongated cells that were mesenchymal in appearance (Figure 1B arrow), 300 μM DETANO treatment led to elongated populations surrounded by cuboidal cells (Figure 1C), and 500 μM appeared to cause higher stress (Figure 1D). As elongation can be characteristic of a migrating cell, scratch test assays were performed to examine the migration potential of NO-cultured cells, which revealed increased migration and gap closure that occurred only in cells that had been maintained in 100 μM DETANO treated cells (Figure 2). Additional in vitro tests for tumorigenesis include cellular ability to grow/survive in reduced serum or serum-free media. Because the 100 μM treated cells migrated faster in the scratch test assay, we evaluated their ability to survive/proliferate in serum-free medium supplemented with selenium, insulin, and transferrin (SIT media). When compared to the untreated control, 100 μM DETANO treated cells formed colonies in SIT media as shown in Appendix A. Three distinct colonies were ring isolated and the remaining cells were pooled for further analysis. Next, control, 100 μM NO-treated parent, and the selected colonies (SIT 2-1, 2-2, 2-3) and pooled (SIT 2-P) cells that grew in SIT media were seeded in soft agar for assessment of their anchorage-independent growth. When compared to control untreated cells, Figure 3A demonstrates altered morphology of SIT 2-1 colonies that grew in soft agar. Epithelial cell growth in three-dimensional culture systems allows formation of acini-like spheroids with hollow lumen, which is apparent in the MCF10A control cells (Figure 3A black arrow) [25]. In contrast, this organized characteristic was not observed in colonies from cells chronically cultured in 100 μM DETANO, which appeared clustered and disorganized (Figure 3A white arrows). Moreover, when compared to untreated control and the 100 μM DETANO treated parent, SIT selected cells formed significantly more colonies (Figure 3B), particularly at lower plating densities (data not shown), which is suggestive of tumorigenesis and stemness [26].

### 3.2. Cytogenetic Alterations

Given the altered morphologies described above, the control and NO treated cells were karyotyped for potential chromosomal changes, which are summarized in Table 1 and Appendix A. Karyotyped untreated cells had 47,XX,dic(1;1)(q11;q11),+der(1),-1,-3,t(3;9)(p13;p22),-5i-8,i(8)(q11;q11),-9,t(9;3;5), characterized by a dicentric Chr 1 composed of two Chr 1 long arm region 1, band 1; a derivative Chr 1; a translocation between Chr 3 short arm region 1, band 3 and Chr 9 short arm region 2, band 2; an isochromosome 8 formed by two Chr 8 long arm region 1, band 1; a translocation involving partial Chr 9 Chr 3 & Chr 5. Karyotype analysis of MCF10A cells maintained for three weeks in 100 μM DETANO had 47,XX,dic(1;1)(q11;q11),+der(1),t(3;9)(p13;p22),der(8),i(8)(q11;q11),t(5;3;9). Advanced spectral karyotyping (SKY) technique was also used to characterize cytogenetic alterations (Appendix A). The SKY assay revealed that control untreated cells carried a deletion on Chr 1; a gain caused by isochromosome 1 long arm; a deletion on Chr 3; a gain on Chr 8, a derivative Chr 9 formed by translocations involved Chr 5, 3 & 9 (Appendix A). Cells maintained in 100 μM NO had a derivative Chr 1 formed by translocation of Chr 1 & Chr 11; a gain of Chr 1 formed by dicentric Chr 1 long arms; a derivative Chr 9 caused by translocation of Chr 9 & Chr 3; a derivative Chr 8; a derivative Chr 9 formed by translocation of Chr 5, 3 & 9.

### 3.3. Mutation Detection in Hot Spot Regions of TP53, PI3KCA, and KRAS

Genomic DNA extracted from control and DETANO treated MCF10A cells was examined for mutations in hot spots regions of three target genes (Figure 4 and Appendix A). Hot spot regions in exons 5–8 of the tumor suppressor p53 (TP53) have been reported [27]. Twenty-three mutations caused by NO-exposure in TP53 gene (did not include the overlapped mutations in control clones) were identified; and among those mutations, 5 were transversion, 10 were transition, and 8 were substitution mutations (Figure 4A and Appendix A). Moreover, the mutations R249T, M237K, T230S, Y163C, and V143L on the TP53 genes were all capable of creating small structural alterations in the gene, hence changing the conformation and binding affinity of TP53. To our surprise, the commonly detected oncogenic TP53 mutant Y220C is replaced with the Y220D mutation in our study [28]. Mutations in KRAS mutation hotspots (exon 1–2) were also examined, where 5 (all transition) mutations were identified, and none of the 5 mutations detected in the control were found in the NO-treated clones (Figure 4B). PIK3CA mutational hotspots (exon 9 & 20) was examined, 5 NO-induced mutations (1 transversion, 1 transition, and 3 substitution) were detected. (Figure 4C and Appendix A). The mutation analysis herein suggests that chronic NO exposure leads to more severe hits or damage to the TP53 gene. To better understand the mutation frequencies in our target genes compared to normal mutation rates, a mutation frequency graph was generated (Figure 4D). The data were normalized by plotting the mutation frequency per 1000 bp and compared to three other genes (FHIT, FGF14, KCNIP4) with relatively large intron size to represent the background mutation rate. This analysis shows increased TP53 genomic alteration frequencies that are dose-dependent with respect to NO concentration when compared to the background mutation rate. In contrast, the NO treatment reduced KRAS mutation frequency, indicating a potential protective effect (Figure 4D).

### 3.4. Cancer Stem Cell Marker Expression

Numerous cancer stem cell (CSC) markers have been identified and characterized in various solid tumors in the past decade [29]. An improved understanding of cancer stemness can provide mechanistic insight regarding tumorigenesis and cancer disease progression that can be exploited for the design of novel therapeutics [30]. Given that increased stem cell biomarker expression was observed in high NOS2 expressing ER- breast tumors [7], we measured CSC biomarker expressions (ALDH1A1, OCT4, CXCR4, CD133, CD117, CD44 and NANOG) in the control and DETANO treated MCF10A cells. Examination of the raw data showed low biomarker expression based upon cycle threshold (Ct) values in untreated control MCF10A cells, which revealed that CD44 transcript (Ct = 16.5) was the most abundant of the CSC markers examined followed by NANOG (Ct = 23.5), OCT4 (Ct = 24.1), CXCR4 (24.5), CD117 (Ct = 30.1), CD133 (Ct = 31.8) and ALDH1A1 (Ct = 33.8). However, after normalizing all data to house-keeping genes, it was found that only CD44 expression was significantly increased following chronic exposure to 300 μM DETANO, when compared to the untreated control (Figure 5). In addition, five NO-induced protein targets, including BRCA1, cMYC, E cadherin, NOS2, and COX2 were examined by western blotting to assess their expression levels in response to chronic NO exposure. Increased COX2 protein expression was observed in 100 and 300 μM treated cells. 

### 3.5. Growth in Nude Mice

Given that the alterations described above are hallmarks of carcinogenesis, the growth potential of NO-treated MCF10A cells in nude mice was assessed next. When compared to untreated control, 100 μM DETANO treated MCF10A parent cells, and the SIT 2-3 clonal population formed masses in nude mice that were significantly larger and persisted for 30 days as shown in Figure 6A. In addition, when grown in the presence of SIT 2-3 clonal cells, MB-468 breast cancer cells formed significantly larger tumors when compared to MB-468 cells alone (Figure 6B).

## 4. Discussion

### 4.1. Chronic NO Exposure-Induced Phenotypic Alterations

The ability to experimentally alter genomic and biochemical pathways have led to the identification of several oncogenes [31]. While malignant transformation of normal cells by the addition of oncogenes (cancer-causing genes) has been effective in rodent cells, human cells have proven resistant to this procedure [31]. NOS2-derived NO has been linked with chronic inflammation, increased TP53 mutation, and elevated cancer risk in patients with inflammatory diseases including Crohn’s disease and inflammatory bowel disease [4,5,32]. In contrast, reduced long-term inflammatory damage to the colon has been identified in NOS2 knockout mice in a trinitrobenzene-induced chronic colitis model [6]. These observations implicate a role of NO in genomic alterations and carcinogenesis. Herein, we extend these earlier findings by showing increased malignant potential in transformed but non-malignant human breast epithelial MCF10A cells following chronic, prolonged exposure to NO. The most obvious changes associated with chronic, prolonged NO exposure included altered morphology where NO treated cells acquired elongated, mesenchymal-like features (Figure 1). Elongated morphology is a characteristic utilized in histopathology and oncology research to describe invasive tumors [33]. Moreover, morphological evaluations of hundreds of tumors vs. normal tissues have allowed discrimination between normal and malignant cells [34]. These cellular morphological characteristics can also be utilized to forecast the condition of cancer cells [34,35]. Thus, the elongation of NO-treated MCF10A cells is consistent with morphological characterizations of malignant transformation [36].

Cancer cells are typically characterized by complex karyotypes including both structural and numerical changes, with aneuploidy being a ubiquitous feature [37]. It is becoming increasingly evident that aneuploidy per se can cause chromosome mis-segregation, which explains the higher rates of chromosome gain/loss observed in aneuploid cancer cells compared to normal diploid cells [37]. These observations prompted us to examine potential chromosomal alterations associated with chronic and prolonged NO treatment. The (3;9) reciprocal translocation (t(3;9)) of MCF10A cells was suggested to be a critical step in the MCF10A cell immortalization process under accelerated growth rate conditions [38]. Cell line behavior regarding both physical and molecular properties can vary between laboratories, which is not uncommon. In our hands, baseline untreated MCF10A cells exhibited extra chromosomes 1 and 5, which is different from initial reports of this cell line [39]. In contrast, NO treated cells had an extra chromosome 1 but not chromosome 5. We cannot rule out possible downstream effects caused by the observed chromosomal alterations. For example, the t(3;9) alteration, also observed in this cell line, was previously reported in association with deletion of the well-known p16/p15 genes, which disrupted cell cycle control [38]. The break points on chromosome 1, 3, 5, 8 and 9 covered a broad range of chromosome lengths and such alterations could play a role in the NO-induced altered phenotypes. Correlations between molecular and cytogenetic alterations induced by NO will require further mapping to identify affected genes.

### 4.2. Chronic NO Exposure-Induced Genomics Alterations

Mutations in TP53 are key drivers of carcinogenesis [40]. Moreover, P53 is a negative regulator of NOS2 [18]. In addition to TP53, PIK3CA and KRAS are also frequently mutated in many tumors. Multiple neoplasms, including head and neck cancer [41], gastric cancer [42], lung cancer [43], ovarian cancer [44], astrocytic tumors [45], breast cancer [46], glioma [47], B cell lymphoma [48], pancreatic cancer [49], and skin cancer have reported mutations within TP53, PIK3CA and KRAS mutation hot spot regions [50]. Given that chromosomal alterations were observed in the NO treated MCF10A cells, the genomic mutation status of TP53, KRAS, and PIK3CA was examined using Sanger sequencing assay [51]. The mutation rate of the TP53 gene, a recognized target of NO, is positively correlated with NO concentration [52]. Mutant hot spot regions on the TP53 gene includes the DNA binding domain, tetramerization domain, and C-terminal regulation domain found in exons 5 to 8 (Figure 4A) [53] and several mutations were identified in exons 5–8 in NO treated MCF10A cells. The significance of these findings can be explained by NO activation of p53, which can have anti-carcinogenic benefits or it can be mutagenic and raise the risk of cancer incidence [18]. NO has been demonstrated to increase p53 accumulation and phosphorylation, namely at ser-15, via ATM and ATR kinases, resulting in G2/M phase cell cycle arrest [4]. In addition, the levels of NOS2 expression and p53-P-ser15 correlated with each other and were favorably linked with the level of inflammation [54]. Consistent with nitrosative stress and the deamination of 5-methylcytosine, p53 mutations were also found in sporadic colon cancer cells and were linked to NOS2 activity in these tissues [18]. These findings uncovered a putative mechanistic relationship between NO and p53 in inflammatory cancer-prone disorders [3,4]. High NOS2 activity (>25 pmol/min/g tissue) in patient lung tissue has been substantially connected with TP53 G:C-to-T:A transversion mutation when compared to lower NOS2 activity in early stage lung cancer [55]. In breast cancer, women who carry germline TP53 mutations have up to 85% increased risk of breast cancer by age 60. Approximately 5–8% of women diagnosed with breast cancer under age 30 had germline TP53 gene mutation [56]. Importantly, tumor NOS2 expression and P53 mutation positively correlated in aggressive ER- breast tumors [7].

On the KRAS gene, the effector domain is found in exons 1 and 2, which are also mutational hotspots, and it has been shown that mutations in the KRAS effector domain can alter tumor progression. [57] In addition, it has been observed that IL1β-induced NOS2 expression greatly enhances the KRAS activating mutation via the activation of promoters on NF-κβ, C/EBP, and CRE-like sites, indicating that NO generation by NOS2 is involved in the tumor-promoting actions of activated KRAS [58]. The byproduct of nitrative DNA damage, 8-nitroguanine, co-localized with NOS2, NOS3, NF-κβ, IKK, MAPK, and MEK, as well as mutant KRAS, indicating that oncogenic KRAS induces extra DNA damage via a signaling cascade including these molecules [59]. Interestingly, a study exploring the connections between KRAS mutations and high/low KRAS expression produced unexpected results, where no difference in KRAS mutation between KRAS-high and KRAS-low individuals was observed. Despite its role in boosting cell proliferation, KRAS-high patients with TNBC exhibited considerably higher Disease-Free Survival and Overall Survival. KRAS-positive TNBC was associated with a favorable tumor immune milieu characterized by increased B cells and CD8^+^ T cells, monocytes, or M1 macrophage. It was also associated with a reduction in CD4^+^ central memory T-cells, but not regulatory T-cells or M2 macrophages [60]. To our surprise, the KRAS gene mutation spectrum in our study exhibits a dose-dependent negative association with NO concentration as the control mutation rate is higher in these hotspots when compared to cells chronically exposed to NO. Therefore, the mutation spectrum for the KRAS gene suggests a protective impact of NO, corroborating the previously mentioned patient study [60].

### 4.3. Chronic NO Exposure-Induced CD44 Stemness Biomarker Expression

Several important signaling pathways, including JAK & STAT, WNT and β-catenin, NANOG and NOTCH modulate the cancer stemness [61,62,63,64,65], which can be gained during dedifferentiation and may promote tumor aggressiveness including the potential for tumor metastasis and regeneration [66,67]. CD44 is a common marker of breast cancer stemness, which is capable of promoting a variety of functions independently or in cooperation with other cell-surface receptors via activation of diverse signaling pathways, including Rho GTPases, Ras-MAPK, and PI3K/AKT pathways, to regulate cell adhesion, migration, survival, invasion, and epithelial-mesenchymal transition (EMT) [68,69]. CD44 mRNA expression was induced following chronic exposure to 300 μM DETANO (Figure 5A) and is consistent with induced expression in high NOS2 expressing ER- breast tumors and NO-induced CD44 protein expression in TNBC breast cancer cells [7,14]. Elevated tumor COX2 expression also regulates stemness and predicts poor survival in ER- breast cancer [70,71,72,73,74]. Moreover, elevated tumor NOS2/COX2 co-expression promotes disease progression and reduced disease-specific survival in ER- breast cancer [75]. Herein, COX2 expression was induced in MCF10A cells treated with 100–300 μM DETANO. In addition, selected populations of these NO-exposed cells formed masses in nude mice that persisted for 30 days.

In conclusion, these results show that chronic, prolonged NO exposure promotes malignant characteristics including altered morphology, chromosomal and genomic instability, increased TP53 mutation, as well as the induction of stemness signaling mediators including CD44 and COX2 in human MCF10A breast epithelial cells, which may in part explain increased cancer risks in patients with inflammatory diseases [4,5].

## Figures and Tables

**Figure 1 biomolecules-13-00311-f001:**
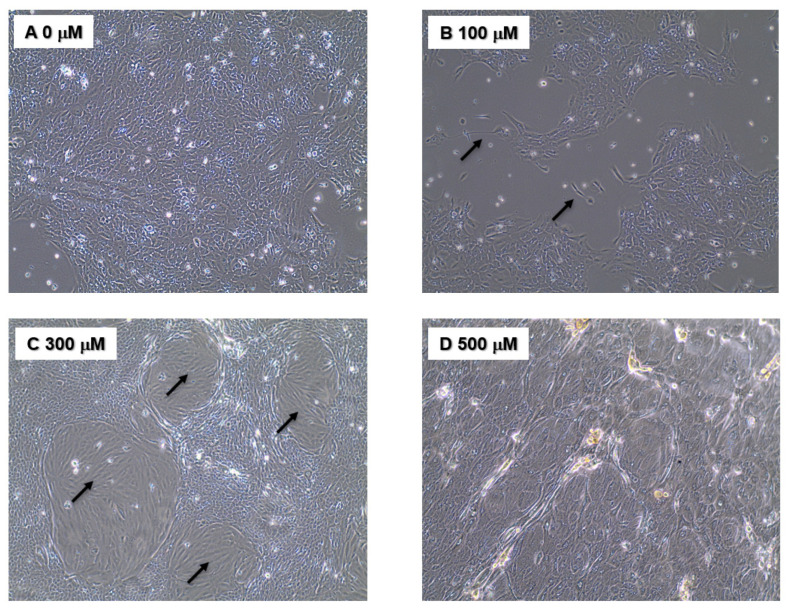
Morphological changes observed in MCF10A cells after prolonged, chronic exposure to NO containing. MCF10A cells were plated at a density of 4 × 10^5^ per 60 mm dish and were treated with 0 (**A**), 100 (**B**), 300 (**C**), or 500 (**D**) μM DETANO twice weekly for three weeks, and then the DETANO was removed, and the cells were analyzed. Elongated cells (black arrow) were observed in NO treated cells.

**Figure 2 biomolecules-13-00311-f002:**
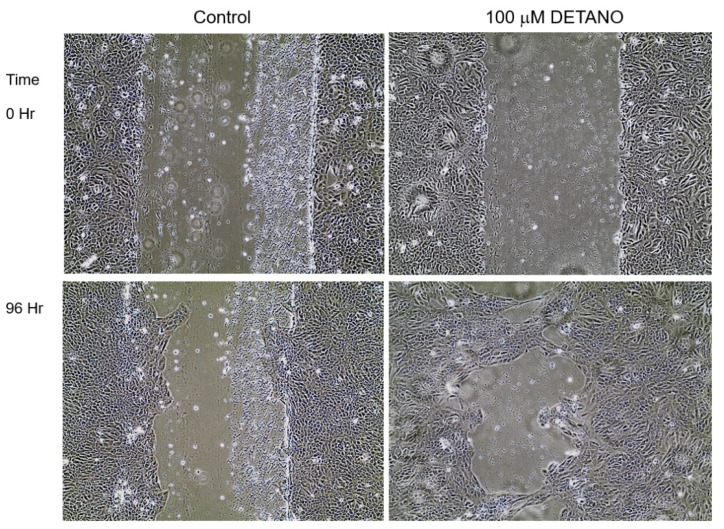
Scratch Test. NO treatment was withdrawn and MCF10A cells were seeded in 60 mm dish and grown to confluence in complete media without treatment. Upon reaching confluence, a straight line was scratched across each confluent monolayer using a 1000 μL pipette tip and the cells were allowed to close the gap in the absence of NO treatment. Top panels show the Control (**left**) as well as cells that had been maintained in 100 μM DETANO (**right**) treated MCF10A cells at time 0. The bottom panels show the same cells at 96 h; when compared to the untreated control, MCF10A cells that had been maintained in 100 μM DETANO closed the gap faster.

**Figure 3 biomolecules-13-00311-f003:**
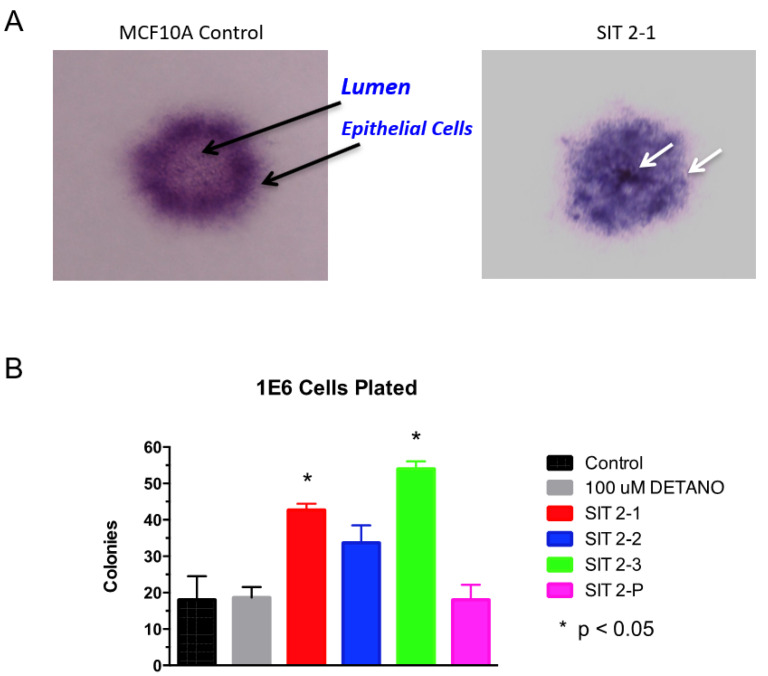
Anchorage Independent Growth in Soft Agar. (**A**) 6-well cell culture plates were prepared with a bottom layer consisting of 0.5% agarose in SIT medium with 5% FBS added. The top layer included 50,000 cells suspended in 3 mL of 0.3% agarose in SIT medium. Black arrows show an acini-like spheroid with hollow lumen formed by control MCF10A cells. White arrows show a clustered and disorganized spheroid formed by MCF10A cells previously maintained in 100 μM DETANO. (**B**) * *p* < 0.05 significantly increased colony formation in soft agar, when compared to untreated control.

**Figure 4 biomolecules-13-00311-f004:**
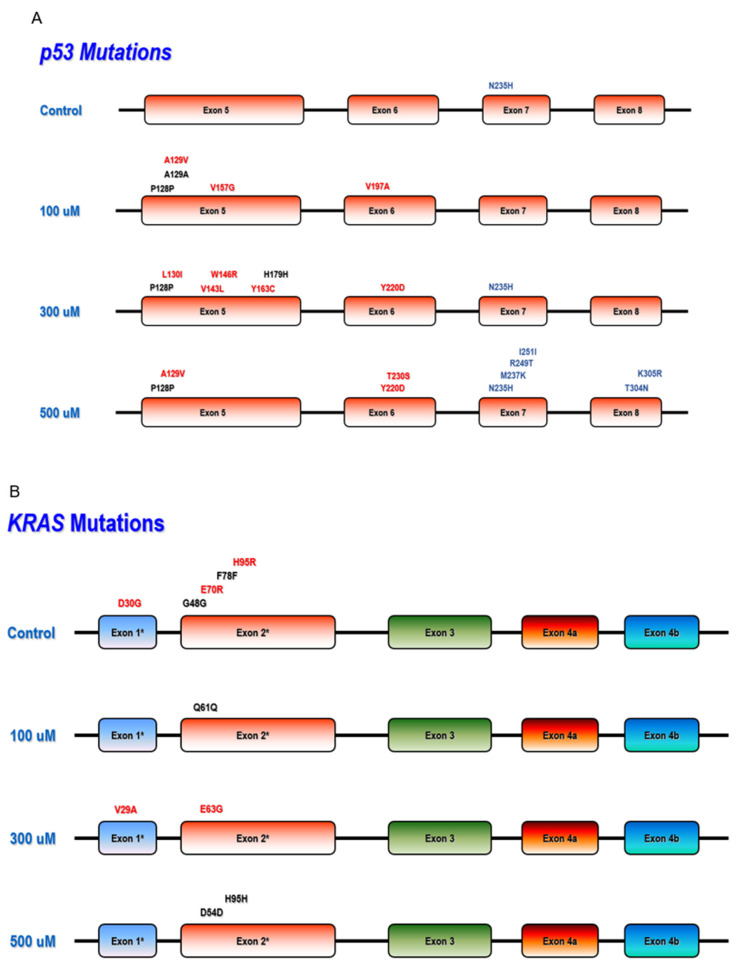
Mutation Hot Spots in TP53, KRAS and PIK3CA genes. Schematic diagrams of (**A**) TP53 (exons 5–8), (**B**) KRAS (exons 1 & 2, indicated with an asterisk), and (**C**) PIK3CA (exon 9 & 20) gene mutation hot spots. Specific mutations identified are shown on top of the gene for different treatment groups. Reported mutations are shown in black font; new mutations are shown in red font, substitution mutation in blue font. (**D**) Mutation Frequency Assessment. The bar charts represent trends of the total number of mutations per 1000 bp, hence no error-bar or statistical significance were provided.

**Figure 5 biomolecules-13-00311-f005:**
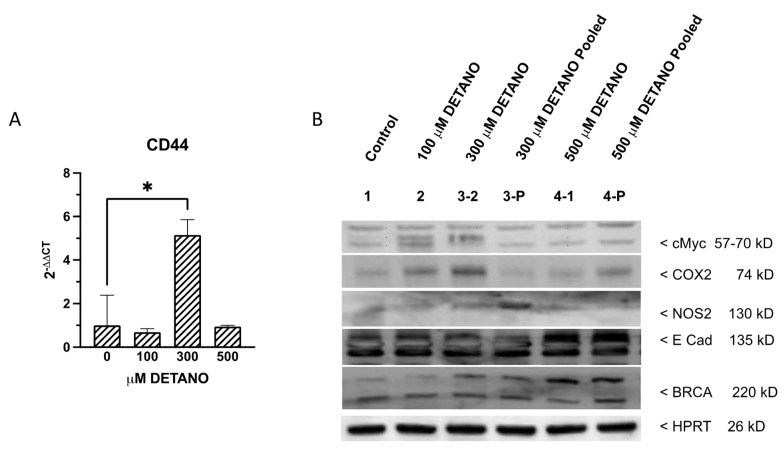
Quantitative Real-Time PCR Results of MCF10A cells following three weeks exposure to 0, 100, 300 or 500 μM DETANO. (**A**) Chronic exposure to 300 μM DETANO led to significantly increased CD44 expression, when compared to the untreated control. (**B**) Increased COX2 protein expression was observed in 100 and 300 μM treated cells. Student t-test was used to determine statistical significance as defined by * *p* < 0.05.

**Figure 6 biomolecules-13-00311-f006:**
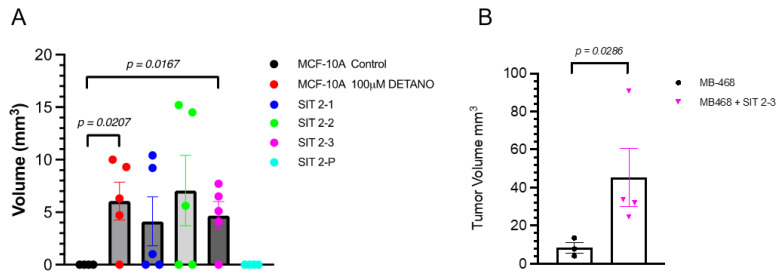
Assessment of the malignant growth potential of subclones in nude mice. (**A**) 100 μM DETANO treated MCF10A parent cells, and the SIT 2-3 clonal population formed small masses in nude mice that were significantly larger than control untreated MCF10A cells. (**B**) When compared to MB-468 cells alone, MB-468 cells injected with SIT 2-3 cells formed significantly larger tumors. Mice were euthanized at 30 days post-cell injection.

**Table 1 biomolecules-13-00311-t001:** Karyotype Analysis Summary.

Control: 47,XX,	dic(1;1)(q11;q11),	+der(1),-1,-3,	t(3;9)(p13;22),-5	del(5),-8,	i(8)(q11;q11),	-9, t(9;3;5)
100 μM: 47,XX,	dic(1;1)(q11;q11),	+der(1),	t(3;9)(p13;p22),	der(8),	i(8)(q11;q11),	t(5;3;9)
2-1 100 μM: 47,XX,	+dic(1q;1q),	der(1) t(1;11),	der(3)t(9;3),	der(8),	der(9)	t(5;3;9)
2-2 100 μM: 47,XX,		der(1),+(1q),	der(3),	der(8),		t(5;3;9)
2-3 100 μM: 47,XX,	+dic(1q;1q),	der(1),	der(3)t(9;3),	der(8),	der(9)	t(5;3;9)

## Data Availability

Whole genome sequencing data is available upon request. Please address all requests to the correspondence author.

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
