# Peer review of "Chronic Exposure to Nitric Oxide Induces P53 Mutations and Malignant-like Features in Human Breast Epithelial Cells"

_biomolecules, 2023, doi:10.3390/biom13020311_

Round 1

Reviewer 1 Report

Robert YS Cheng and colleagues submitted a well-written experimental manuscript focused on chronic exposure to nitric oxide induces p53 mutations and malignant-like features in human breast epithelial cells.

Authors investigated the possible effects of chronic NO exposure on MCF10A breast epithelial cells, as defined by changes in cellular morphology, chromosome/genomic stability, RNA and protein expression, and altered cell phenotypes. For that human MCF10A cells were maintained in varying doses of the NO donor DETANO for three weeks. Distinct patterns of genomic modifications in TP53 and KRAS target genes were detected in NO-treated cells when compared to background mutations. In addition, quantitative real-time PCR demonstrated an increase in the expression of cancer stem cell marker CD44 after prolonged exposure to 300 uM DETANO. While similar changes in cell morphology were found in cells exposed to 300-500 uM DETANO, cells cultured in 100 uM DETANO exhibited enhanced motility. In addition, 100 uM NO-treated cells proliferated in serum-free media and selected clonal populations and pooled cells formed colonies in soft agar that were clustered and disorganized.

Finally, authors conclude that they demonstrated that chronic, prolonged NO exposure promotes malignant characteristics including altered morphology, chromosomal and genomic instability, increased TP53 mutation, as well as the induction of stemness signaling mediators including CD44 and COX2 in human MCF10A breast epithelial cells, which may in part explain increased cancer risks in patients with inflammatory diseases

Overall, authors present a quality and well-written manuscript valuable for the scientific community and should be accepted for publication after minor edits are made.

===============

Other comments:

1) Please check for typos and punctuation.

2) With regards to p53 mutations authors are kindly encouraged to cite the following article that describes the development of novel activators of mutant p53. DOI: 10.1021/acsptsci.2c00164

Author Response

We appreciate Reviewer #1's enthusiastic remarks and suggestions.

Comment: 1) Please check for typos and punctuation.

Response: We have proofread the manuscript for spelling, grammatical, and punctuation errors with Microsoft Editor and made the appropriate changes.

Comment: 2) With regards to p53 mutations authors are kindly encouraged to cite the following article that describes the development of novel activators of mutant p53. DOI: 10.1021/acsptsci.2c00164.

Response: We have added the following statement which references the suggested publication to the text. “To our surprise, the commonly detected oncogenic TP53 mutant Y220C is replaced with the Y220D mutation in our study.” This amendment is highlighted in yellow font in section 3.3 on p10.

Reviewer 2 Report

In this article the authors look at NO exposure using the chemical DETA/NO and examine to what extent it is mutagenic in cell lines.

Major comments:

1. Figure 3 is not contributing much to the article. However, Sup 2 is interesting and showing how different CSC markers are differentially expressed in NO exposed cells. I would like to suggest to make Figure 3 supplemental and sup2 a main figure

2. The text on page 11 mentions that 5 protein targets of NO were examined. Only COX2 expression is shown, but none of the others are shown. Please add.

Author Response

We appreciate the insightful comments and recommendations from Reviewer 2.

Comment: 1. Figure 3 is not contributing much to the article. However, Sup 2 is interesting and showing how different CSC markers are differentially expressed in NO exposed cells. I would like to suggest making Figure 3 supplemental and sup2 a main figure.

Response: We have relocated Figure 3 to the Supplemental section, which is now Supplemental Figure 1. However, we wish to withdraw the original Supplemental Figure 2 from the paper because this figure reflects data that was not normalized to the house-keeping gene. Therefore, when normalized to the house-keeping gene, only CD44 was significantly elevated in 300 mM treated cells. This finding is consistent with our earlier publications that showed significant inductions of CD44 in high NOS2 expressing ER- tumors (1) while MB231 xenografts in mice treated with the NOS2 inhibitor aminoguanidine demonstrated reduced CD44 expression when compared to xenografts from untreated control mice (2). We apologize for this lack of clarity. We have amended to text (highlighted in yellow) in section 3.4 p.13 as follows;

3.4.      Cancer Stem Cell Marker Expression

Numerous cancer stem cell (CSC) markers have been identified and characterized in various solid tumors in the past decade (3). An improved understanding of cancer stemness can provide mechanistic insight regarding tumorigenesis and cancer disease progression that can be exploited for the design of novel therapeutics (4). Given that increased stem cell biomarker expression was observed in high NOS2 expressing ER- breast tumors (1), we measured CSC biomarker expressions (ALDH1A1, OCT4, CXCR4, CD133, CD117, CD44 and NANOG) in the control and DETANO treated MCF10A cells. Examination of the raw data showed low biomarker expression based upon cycle threshold (Ct) values in untreated control MCF10A cells, which revealed that CD44 transcript (Ct=16.5) was the most abundant of the CSC markers examined followed by NANOG (Ct=23.5), OCT4 (Ct=24.1), CXCR4 (24.5), CD117 (Ct=30.1), CD133 (Ct=31.8) and ALDH1A1 (Ct=33.8). However, after normalizing all data to house-keeping genes, it was found that only CD44 expression was significantly increased following chronic exposure to 300 mM DETANO, when compared to the untreated control (Figure 6).  In addition, five NO-induced protein targets, including BRCA1, cMYC, E cadherin, NOS2, and COX2 were examined by western blotting to assess their expression levels in response to chronic NO exposure. Increased COX2 protein expression was observed in 100 and 300 mM treated cells.

Comment: 2. The text on page 11 mentions that 5 protein targets of NO were examined. Only COX2 expression is shown, but none of the others are shown. Please add.

Response: We have added the additional data as requested, which is now shown in Figure 5.

  1. Glynn, S. A., Boersma, B. J., Dorsey, T. H., Yi, M., Yfantis, H. G., Ridnour, L. A., Martin, D. N., Switzer, C. H., Hudson, R. S., Wink, D. A., Lee, D. H., Stephens, R. M., and Ambs, S. (2010) Increased NOS2 predicts poor survival in estrogen receptor-negative breast cancer patients. J Clin Invest 120, 3843-3854
  2. Heinecke, J. L., Ridnour, L. A., Cheng, R. Y., Switzer, C. H., Lizardo, M. M., Khanna, C., Glynn, S. A., Hussain, S. P., Young, H. A., Ambs, S., and Wink, D. A. (2014) Tumor microenvironment-based feed-forward regulation of NOS2 in breast cancer progression. Proc Natl Acad Sci U S A 111, 6323-6328
  3. Mayani, H., Chavez-Gonzalez, A., Vazquez-Santillan, K., Contreras, J., and Guzman, M. L. (2022) Cancer Stem Cells: Biology and Therapeutic Implications. Arch Med Res 53, 770-784
  4. Kim, W. T., and Ryu, C. J. (2017) Cancer stem cell surface markers on normal stem cells. BMB reports 50, 285-298

Round 2

Reviewer 2 Report

The new data contributes to the manuscript and all other points have been addressed sufficiently